# High MB Solution Degradation Efficiency of FeSiBZr Amorphous Ribbon with Surface Tunnels

**DOI:** 10.3390/ma13173694

**Published:** 2020-08-21

**Authors:** Qi Chen, Zhicheng Yan, Hao Zhang, Lai-Chang Zhang, Haijian Ma, Wenlong Wang, Weimin Wang

**Affiliations:** 1Key Laboratory for Liquid-Solid Structural Evolution and Processing of Materials (Ministry of Education), School of Materials Science and Engineering, Shandong University, Jinan 250061, China; caesar@mail.sdu.edu.cn (Q.C.); rengaryzc@outlook.com (Z.Y.); zhanghao_0611@163.com (H.Z.); 2School of Engineering, Edith Cowan University, 270 Joondalup Drive, Joondalup, Perth WA6027, Australia; l.zhang@ecu.edu.au; 3School of Mechanical, Electrical and Vehicle Engineering, Weifang Institute, Weifang 261061, China; hjma@wfu.edu.cn; 4National Engineering Laboratory for Reducing Emissions from Coal Combustion, Engineering Research Center of Environmental Thermal Technology of Ministry of Education, Shandong Key Laboratory of Energy Carbon Reduction and Resource Utilization, School of Energy and Power Engineering, Shandong University, Jinan 250061, China; wwenlong@sdu.edu.cn

**Keywords:** Fe-based alloys, methylene blue dye, free volume, surface tunnel

## Abstract

The as spun amorphous (Fe_78_Si_9_B_13_)_99.5_Zr_0.5_ (Zr0.5) and (Fe_78_Si_9_B_13_)_99_Zr_1_ (Zr1) ribbons having a Fenton-like reaction are proved to bear a good degradation performance in organic dye wastewater treatment for the first time by evaluating their degradation efficiency in methylene blue (MB) solution. Compared to the widely studied (Fe_78_Si_9_B_13_)_100_Zr_0_ (Zr0) amorphous ribbon for degradation, with increasing *c*_Zr_ (Zr atomic content), the as-spun Zr0, Zr0.5 and Zr1 amorphous ribbons have gradually increased degradation rate of MB solution. According to $\delta_{c}$ (characteristic distance) of as-spun Zr0, Zr0.5 and Zr1 ribbons, the free volume in Zr1 ribbon is higher Zr0 and Zr0.5 ribbons. In the reaction process, the Zr1 ribbon surface formed the 3D nano-porous structure with specific surface area higher than the cotton floc structure formed by Zr0 ribbon and coarse porous structure formed by Zr0.5 ribbon. The Zr1 ribbon’s high free volume and high specific surface area make its degradation rate of MB solution higher than that of Zr0 and Zr0.5 ribbons. This work not only provides a new method to remedying the organic dyes wastewater with high efficiency and low-cost, but also improves an application prospect of Fe-based glassy alloys.

## 1. Introduction

Nowadays, organic dyes wastewater is commonly produced in the industrial production of dyestuff, textiles, paper and plastics. This organic dyes wastewater contains carcinogenic, teratogenic and biological toxic substances, which will cause serious pollution to the environment [1,2,3]. Thus, increasing attention has been paid to the harmful pollution of industrial organic dyes to our water bodies. In the past several decades, numerous research works have been done to reduce their hazardous effects, including the physical adsorption of activated carbon and clay [4,5], biodegradation [6], chemical degradation by advanced oxidation process [7,8,9,10] and degradation of specific alloys [11,12,13,14]. However, these methods have obvious disadvantages such as low-efficiency, high-cost and short service life. Thus, we should actively explore advanced materials to better degrade organic dyes in polluted water [15].

Amorphous alloys (metallic glass alloys) have excellent properties such as high strength and corrosion resistance due to their long-range disordered atomic structures, and the importance of fundamental scientific and engineering application potential is attracting rising attention [16,17,18]. At present, the amorphous alloy ribbons including Mg-based ribbons [19,20,21,22,23], Al-based ribbons [24,25,26], Co-based ribbons [27,28,29] and Fe-based ribbons [30,31,32,33,34,35,36] have been proved to have good degradation properties to organic dyes wastewater. It is well-believed that the excellent degradation ability of amorphous alloys is due to three factors: (1) their high residual stress, (2) thermodynamic instabilityand (3) lots of unsaturated sites on the surface.

Among Fe-, Cu-, Al- and Mg-based amorphous alloys used to degrade organic dyes wastewater, Fe-based alloys have attracted the most attention because of their high degradation efficiency, low cost and good reusability. The frequently used Fe-based alloys for degradation are FeSiB systems, and FeSiB alloys with other elements. Jia et al., using the prepared Fe_78_Si_9_B_13_ and Fe_73.5_Si_13.5_B_9_Cu_1_Nb_3_ amorphous ribbons, showed higher degradation efficiency when degrading cibacron brilliant red 3B-A, methyl blue and methyl orange dyes by Fenton-like reaction (Fe^0^ + H_2_O_2_ → Fe^2+^ + 2OH^−^, Fe^2+^ + H_2_O_2_ → Fe^3+^ + •OH + OH^−^, •OH + organics → products), which proved that Fe-based amorphous ribbons have better degradation performance of organic dyes than other kinds of amorphous alloys [37,38]. Xie et al. used amorphous Fe_76_Si_9_B_12_Y_3_ powder to degrade methyl orange dye, which was 1000 times more reactive than industrial iron powder [39].

In this paper, using (Fe_78_Si_9_B_13_)_99.5_Zr_0.5_ (Zr0.5) and (Fe_78_Si_9_B_13_)_99_Zr_1_ (Zr1) amorphous ribbons, the methylene blue (MB) dyes degradation by Fenton-like reactions is reported for the first time, and is compared with Fe_78_Si_9_B_13_ (Zr0) amorphous ribbon. The addition of trace Zr element effectively adjusts the interatomic force of Zr0 amorphous ribbon, which makes the atoms on the surface of amorphous ribbon participate in Fenton-like reaction more easily, and with the increase of Zr element content, it is easier to form developed 3D nano-porous acicular structure on the ribbon surface, thus increasing the specific surface area of reaction and improving the degradation rate. The effects of initial pH and H_2_O_2_ concentration on the degradation efficiency of MB using the Zr0, Zr0.5 and Zr1 amorphous ribbons during Fenton-like reactions are investigated. This work not only provides a new routine for the remediation of organic dyes wastewater, but also extends the application range of Fe-based amorphous/glassy alloys.

## 2. Experimental

### 2.1. Materials and Reagent

Alloy ingots with a nominal composition of (Fe_78_Si_9_B_13_)_100_Zr_0_ (Zr0, at.%), (Fe_78_Si_9_B_13_)_99.5_Zr_0.5_ (Zr0.5, at.%) and (Fe_78_Si_9_B_13_)_99_Zr_1_ (Zr1, at.%) were prepared by arc melting of pre-alloyed Fe_78_Si_9_B_13_ ingots and high-purity Zr (99.99 wt.%) in an arc melting furnace (MAM-1 Edmund Buhler, Berlin, Germany), which was vacuumed to 5 × 10^−3^ Pa first and then filled with purified argon (99.999%). The ribbons with thickness of ~30 μm and width of ~2.5 mm were prepared by single copper roller melt-spinning (i.e., planar flow casting) system. The roller/wheel speed was 42 m·s^−1^. The amorphous ribbons were cut into 5 cm long strips for degradation tests. Commercially available methylene blue (MB, C_16_H_18_ClN_3_S, AR grade, Tianjin Beichen Fangzheng Reagent Factory, Tianjin, China), Hydrochloric acid (HCl, AR grade, Sinopharm Chemical Reagent Co., Ltd., Shanghai, China), Sodium hydroxide (NaOH, AR grade, Tianjin Hengxing Chemical Reagent Manufacturing Co., Ltd., Tianjin, China) and Hydrogen peroxide (H_2_O_2_, AR grade, Tianjin Kemeo Chemical Reagent Co., Ltd. Tianjin, China) are used in the experiment.

### 2.2. Characterization

The amorphous structure of the as-spun ribbons (in a stripe-like shape) was investigated by X-ray diffraction (XRD, Bruker D8 Discover, Brooke (Beijing) Technology Co., Ltd., Beijing, China) with Cu-Kα radiation and transmission electron microscopy (TEM, FEI Tecnai G2 F20, American FEI Company, Portland, OR, America). The amorphous character of the ribbon samples was also confirmed by differential scanning calorimetry (DSC, Netzsch-404, Netzsch, Bavaria, Germany) at a heating rate of 20 K/min. The surface morphology of the as-spun and reacted ribbons was observed by scanning electron microscope (SEM, JSM-7800F, Japan Electronics Co., Ltd., Beijing, China) equipped with an energy dispersive X-ray spectrometer (EDS).

### 2.3. Degradation Tests

Preparing MB solution in 500 mL volumetric flask with deionized water (DW), then pour 100 mL MB solution (100 mg L^−1^) in 250 mL beakers. A certain mass of ribbons (0.5 g L^−1^) and H_2_O_2_ (1 mM) was placed to the MB solution, stirring at a fixed speed (300 r min^−1^) during the degradation process, and the constant temperature (298 K) of the MB solution was maintained with a water bath. The pH (pH = 3) of the MB solution was adjusted using 12 mol L^−1^ HCl, as well as 1 M NaOH. At time intervals, a 3 mL solution was extracted with a syringe and filtered with a 0.45 μm membrane, and the concentration of MB solution was monitored in real-time with UV-Vis spectrophotometer (UV-4802) to obtain the absorbance spectrum of the solution.

### 2.4. Electrochemical Tests

The polarization curves and electrochemical impedance spectra (EIS) were measured using an electrochemical measuring instrument (CHI 660E, Shanghai Chenhua Instrument Co., Ltd., Shanghai, China) in the 20 mL DW or MB solutions (pH = 3, *T* = 298 K, *C*_H2O2_ = 1 mM and *C*_MB_ = 100 mg·L^−1^). The three-electrode cell was used for measurement, the saturated calomel electrode (SCE) was used as reference electrode, platinum was used as counter electrode and as-spun ribbon was used as working electrode. When the open-circuit potential stabilizes, the polarization curve was recorded at the potential scanning speed of 1 mV s^−1^. EIS was performed in static states, with scanning frequencies from 100 kHz to 0.01 Hz and the amplitude of ± 10 mV.

## 3. Results

### 3.1. XRD, DSC and TEM Analysis

Figure 1a shows the x-ray diffraction XRD curves of as-spun (Fe_78_Si_9_B_13_)_100_Zr_0_ (Zr0), (Fe_78_Si_9_B_13_)_99.5_Zr_0.5_ (Zr0.5) and (Fe_78_Si_9_B_13_)_99_Zr_1_ (Zr1) ribbons. The XRD curves of as-spun Zr0, Zr0.5 and Zr1 ribbons have only a typical diffuse scattering peak at 2*θ*_max_ = 44.45°, 44.32° and 44.14° respectively, indicating that the as-spun Zr0, Zr0.5 and Zr1 ribbons owns a fully amorphous structure. With the increase of Zr content, the diffuse scattering peak moves to a low degree. The mean neighboring atomic distance increases with Zr content due to the large atom size of Zr element [40]. Figure 1b shows the DSC curves of as-spun Zr0, Zr0.5 and Zr1 ribbons. In the DSC curves of as-spun Zr0, Zr0.5 and Zr1 ribbons, there are two crystallization peak temperatures *T*_P1_ and *T*_P2_, a melting peak temperature *T*_P3_ in the heating scan, respectively. The *T*_P2_-*T*_P1_ (crystallization peak temperature range) of as-spun Zr0, Zr0.5 and Zr1 ribbons are 18, 41 and 68 K, respectively. Not only are the two crystallization peak temperatures *T*_P1_ and *T*_P2_ of as-spun Zr0.5 and Zr1 ribbons higher than that of as-spun Zr0 ribbon, but also their crystallization peak temperature range of *T*_P2_-*T*_P1_ higher than that of as-spun Zr0 ribbon, indicating that the as-spun Zr0.5 and Zr1 ribbons have better amorphous stability and formability. The melting peak temperature *T*_P3_ of as-spun Zr0.5 and Zr1 ribbons was lower than that of as-spun Zr0 ribbon, indicating that the addition of Zr element can reduce the bonding force between atoms in amorphous ribbons, thus have a lower melting peak temperature.

In order to further characterize the microstructure of the as-spun Zr0, Zr0.5 and Zr1 ribbons, we conducted TEM investigations, which are shown in Figure 2. There is mainly maze shape pattern without crystallites in the high-resolution bright field images of the as-spun Zr0, Zr0.5 and Zr1 ribbons (Figure 2a–c), and the corresponding SAED (Selected area electron diffraction) patterns have two typical diffraction halos (Figure 2d–f), confirming that the as-spun Zr0, Zr0.5 and Zr1 ribbons own a fully amorphous structure. Thus, the results of TEM are agreeing with the XRD curves and DSC curves (Figure 1a,b).

### 3.2. Degradation Performance

Figure 3a–c exhibits the ultraviolet–visible (UV–vis) absorption spectra of filtered methylene blue (MB) solution in a series of time intervals after adding the as-spun Zr0, Zr0.5 and Zr1 ribbons in the reaction batch, respectively. The UV–vis absorption spectra of MB solution have two major absorption peaks at about 610 nm and 654 nm, which represent the auxochrome and chromophore groups, respectively [38]. The normalized concentration of the MB solution obtains its peak value at 654 nm, which represents the chromogenic species, as shown in Figure 3d. With increasing *t*_r_ the absorption peak at 654 nm gradually decayed, indicating that the chromophore groups of MB disappeared gradually. In the first 9 min, the auxochrome groups react more quickly with Fe-based ribbons than the chromophore groups. The degradation kinetics is usually described by the pseudo-first-order equation as follows [41]:*C_t_* = *C*_0_ exp(−*kt*_r_),(1)
where *k* is the reaction rate constant (min^−1^), *C*_0_ is the initial concentration of MB solution (mg L^−1^), *t*_r_ is the reaction time (min), and *C_t_* is the instant concentration of MB solution (mg L^−1^) at *t*_r_. In this work, the ln (*C*_0_*/C_t_*) − *t*_r_ curves are shown in the inset of Figure 3d. The deduced *k* of as-spun Zr0.5 and Zr1 ribbons are 0.22 min^–1^ and 0.24 min^–1^, which is larger than 0.19 min^–1^ for as-spun Zr0 ribbon. Here, the fit goodness values *R*^2^ of Zr0, Zr0.5 and Zr1 ribbons are 0.97, 0.98 and 0.97, respectively. Thus, the as-spun Zr0.5 and Zr1 ribbons bear a higher degradation performance for MB solution compared with as-spun Zr0 ribbon.

### 3.3. Surface Morphology

In order to deeply understand the MB solution degradation mechanism with the Zr0, Zr0.5 and Zr1 amorphous ribbons, we test to study the structural evolution of the ribbon surfaces during the Fenton-like reaction process. The SEM images on the surface of the as-spun and reacted Zr0, Zr0.5 and Zr1 ribbons are displayed in Figure 4 and the EDS results are listed in Table 1. The as-spun Zr0, Zr0.5 and Zr1 ribbons have a typical smooth amorphous surface, as shown in Figure 4a–c, respectively. There are cotton floc structures and some corrosion pit on the reacted Zr0 ribbon (Figure 4d). The reacted Zr0.5 ribbon surface appears in coarse porous structure with the ligament width of 100 nm (Figure 4e). The reacted Zr1 ribbon surface has a developed 3D nano-porous structure with some acicular matters on the ligaments (Figure 4f). The porous surface structures should have enhancing effects on the degradation process because they can provide mass transfer channels. Compared with Zr0 and Zr0.5 ribbons, the higher pore density of Zr1 ribbon may be the reason for its higher degradation performance in Fenton-like reaction.

The XRD patterns of Zr0, Zr0.5 and Zr1 ribbons (upper right inset in Figure 4d–f) after reaction still have only a typical diffuse scattering peak at 2*θ*_max_ = 44.56°, 44.47° and 44.39° respectively, indicating that they still own a fully amorphous structure. Here, the different peak positions of the reacted ribbons are higher than the as-spun ribbons. Moreover, the 2*θ*_max_ deviation Δ2*θ*_max_ values of Zr0, Zr0.5 and Zr1 ribbons are 0.11°, 0.15° and 0.25°, respectively, showing a high tendency with increasing *c*_Zr_. After reacting with MB solution for 15 min, the Zr0, Zr0.5 and Zr1 ribbons have a decreased *c*_Fe_, and the *c*_Zr_ has remained basically unchanged. Moreover, the decrease of Fe element on the surface of Zr1 ribbon is higher than that of Zr0 and Zr0.5 ribbons, indicating that the high degradation rate of Zr1 ribbon is due to a large amount of Fe element participating in the Fenton-like reaction. After degradation, the *c*_O_ on Zr0, Zr0.5 and Zr1 ribbon surface increases, indicating that the degradation process involves the oxidation of the ribbons.

### 3.4. Electrochemical Analysis

The polarization curves and electrochemical impedance spectra (EIS) of the as-spun Zr0, Zr0.5 and Zr1 ribbons in DW and MB solution (*T* = 298 K, pH = 3, *C*_H2O2_ = 1 mM and *C*_MB_ = 100 mg L^−1^) are shown in Figure 5. In DW, the corrosion potentials (*E*_corr_) of the Zr0.5 and Zr1 ribbons are −0.75 and −0.71 V (Figure 5a), higher than the Zr0 ribbon (−0.80 V). The corrosion current densities (*i*_corr_) of the Zr0.5 and Zr1 ribbons are 5.83 × 10^−6^ and 3.30 × 10^−6^ A·cm^−2^, lower than the Zr0 ribbon (8.23 × 10^−6^ A cm^−2^). In MB solution, the *E*_corr_ of the Zr0.5 and Zr1 ribbons are −0.60 and −0.54 V (Figure 5b), higher than the Zr0 ribbon (−0.65 V). The *i*_corr_ of the Zr0.5 and Zr1 ribbons are 1.54 × 10^−4^ and 1.20 × 10^−4^ A cm^−2^, lower than the Zr0 ribbon (1.82 × 10^−4^ A cm^−2^). The above data from polarization curves indicate that the Zr0.5 and Zr1 ribbons have better corrosion resistance than the Zr0 ribbon in DW and MB solution.

In both DW and MB solution (Figure 5c,d), the Nyquist semicircle diameter of the Zr0.5 and Zr1 ribbons is larger than that of Zr0 ribbon. The equivalent circuit composed of R(Q(R(QR))) is used to fit the EIS data. The fitting error (chi square χ^2^) for the Zr0, Zr0.5 and Zr1 ribbons are 1.47 × 10^−5^, 1.07 × 10^−4^ and 6.54 × 10^−4^ in DW, respectively; while they are 1.05 × 10^−5^, 6.77 × 10^−6^ and 7.71 × 10^−6^ in MB solution, respectively. In the equivalent circuit, the phase element (CPE) *Q* is defined as [42]:*Q* = (*jw*)^−n^/*Y*_0_,(2)
where *Q* is the resistance, *j* is the imaginary unit, *w* is the frequency, n is the coefficient of CPE and *Y*_0_ is the admittance.

The fitting results are summarized in Table 2. The *R*_s_ (solution resistance) of the Zr0, Zr0.5 and Zr1 ribbons in MB solution is lower than that in DW, which may be due to the conductive H^+^ in MB solution. The *R*_f_ (resistance of passivation film) and *R*_a_ (resistance of electrochemical reaction) of the Zr0, Zr0.5 and Zr1 ribbons in DW are higher than those in MB solution respectively, due to a certain amount of H^+^ and methylene blue molecules in the MB solution. The *R*_total_ (total resistance) of the Zr0, Zr0.5 and Zr1 ribbons in DW is higher than that in MB solution, and the *R*_total_ of Zr0.5 and Zr1 ribbons are higher than that of Zr0 ribbon in either DW or MB solution. Thus, the EIS results are also in good agreement with the results of polarization curves (Figure 5a,b).

### 3.5. Effect of pH on Ribbon Degradation

The working pH range of the Fenton-like reaction using the as-spun Zr0, Zr0.5 and Zr1 ribbons for the MB solution degradation has been studied, and keep other reaction conditions constant: *T* = 298 K, *C*_H2O2_ = 1 mM, ribbon dosage = 0.5 g L^−1^ and *C*_MB_ = 100 mg L^−1^. The highest degradation rate for as-spun Zr0, Zr0.5 and Zr1 ribbons is achieved at pH =3, as shown in Figure 6a−c, respectively. Surprisingly, as pH = 2, the degradation efficiency (*η* = (1 − *C*_t_/*C*_0_ × 100%, *t* = 15 min) of as-spun Zr0, Zr0.5 and Zr1 ribbons is lower than that at pH = 3 (Figure 6d). This may be because the iron in the ribbon dissolved to generate hydrogen when the H^+^ in the solution is too high (H^+^ + Fe^0^ → Fe^2+^ + H_2_ ↑). This hydrogen evolution reaction generates a large amount of Fe^2+^, which may consume •OH and lower the oxidation capability of the MB solution (Fe^2+^ + •OH → OH^−^ + Fe^3+^). As pH > 3, the *η* of the MB solution decreases with the increasing pH, as there must be enough H^+^ in the solution to carry out Fenton-like reactions. When the pH increases to 4, this MB solution hardly degrades before 7 min, and then it degrades slowly. As pH > 5 the *η* of MB solution is nearly zero.

### 3.6. Effect of H_2_O_2_ Concentration on Ribbon Degradation and Surface Morphology

The concentration of H_2_O_2_ controls the rate of •OH generation in the Fenton-like reactions. The effect of the concentration on the degradation process of MB solution using as-spun Zr0, Zr0.5 and Zr1 ribbons was investigated, as shown in Figure 7. Various concentrations of H_2_O_2_, including 0, 0.5, 1, 5, 10, 30 and 50 mM, are added to MB solution, and other reaction conditions constant: *T* = 298 K, pH = 3, ribbon dosage = 0.5 g L^−1^ and *C*_MB_ = 100 mg L^−1^. It is proved that H_2_O_2_ is necessary to degrade MB solution, because the *η* is very low without adding H_2_O_2_ (Figure 7d). When the *C*_H2O2_ is 0.5 mM, the concentration of MB solution remained basically unchanged after 7 min of degradation, which may be due to the fact that H_2_O_2_ was completely consumed and •OH could not be produced in Fenton-like reaction. With the *C*_H2O2_ increasing from 1 to 10 mM, the *η* of Fe-based ribbons increases gradually. When the *C*_H_2_O_2__ increases to 30 and 50 mM, the *η* of MB solution began to decrease significantly. The results show that the appropriate addition of H_2_O_2_ can effectively accelerate the degradation process. However, due to the well-known hydroxyl radical scavenging effect, excessive H_2_O_2_ is not beneficial to the degradation process (H_2_O_2_ + •OH → H_2_O + •HO_2_). The oxidation potential of generated radical •HO_2_ is much lower than that of •OH, which slows the degradation rate of MB solution.

Figure 8 shows the SEM images of as-spun Zr1 ribbon reacted with MB solution at different H_2_O_2_ concentrations. The surface of the as-spun Zr1 ribbon is relatively smooth (Figure 4c), but fuzzy 3D nano-porous structures appear on the surface of the ribbon as *C*_H2O2_ = 0 mM (Figure 8a). With the *C*_H2O2_ increasing from 0.5 to 5 mM, the 3D nano-porous structures develop and coarsen (Figure 4f and Figure 8b,c). As *C*_H_2_O_2__ = 10 mM, the 3D nano-porous structure begins to transform into an intersecting grid-like structure (Figure 8d). When the *C*_H_2_O_2__ reaches 30 and 50 mM, the intersecting grid-like structure gradually becomes thicker and more developed (Figure 8e,f). Table 3 summarizes the EDS results of the Zr1 ribbon reacted with MB solution at different H_2_O_2_ concentrations. Comparing the EDS results of the as-spun and reacted Zr1 ribbons (*C*_H2O2_ = 1 mM, Table 1), the Fe content decrement on Zr1 ribbon surface increases gradually with increasing *C*_H_2_O_2__, the *c*_Si_ and *c*_B_ have remained basically unchanged simultaneously. However, the *c*_O_ rise on the surface of the Zr1 ribbons increases gradually, which is due to H_2_O_2_ having strong oxidizability. Excessive H_2_O_2_ will oxidize the metal on the ribbon surface, resulting in a large number of oxides.

## 4. Discussion

With increasing *c*_Zr_, the 2*θ*_max_ of Fe-based ribbons decreases gradually. Based on the element date [43], the average atomic radius $\bar{r}$ of Zr0, Zr0.5 and Zr1 ribbons is 1.301, 1.302 and 1.303 Å, respectively. According to the equation on the characteristic distance [44,45,46]:(3)$$\delta_{c}=\frac{2\pi }{Q_{P}},\;\mathrm{with}\;Q_{P}=(4\pi \;\sin\theta_{\max})/\lambda ,$$
where $\delta_{c}$ is the characteristic distance, $Q_{P}$ is the diffraction vector of the diffusive maximum of the XRD pattern, and the λ is the X-ray wavelength (Cu-Kα = 1.5418 Å). The $\delta_{c}$ value of as-spun Zr0, Zr0.5 and Zr1 ribbons is 2.036, 2.042 and 2.050 Å, respectively. Here, the $\frac{{\bar{r}}_{\mathrm{Zr}0.5}}{{\bar{r}}_{\mathrm{Zr}0}}$ and $\frac{{\bar{r}}_{\mathrm{Zr}1}}{{\bar{r}}_{\mathrm{Zr}0}}$ equal 1.00077 and 1.00154; meanwhile, $\frac{\delta_{c,\mathrm{Zr}0.5}}{\delta_{c,\mathrm{Zr}0}}$ and $\frac{\delta_{c,\mathrm{Zr}1}}{\delta_{c,\mathrm{Zr}0}}$ equal 1.00295 and 1.00688, which are much higher than the corresponding $\bar{r}$ radius. The high $\frac{\delta_{c,\mathrm{Zr}1}}{\delta_{c,\mathrm{Zr}0}}$ suggest that Zr atom can introduce free volume into the glassy matrix [47]. On the other hand, Zr atom can enhance the stability of Fe-Zr and Fe-B bonds in the liquid state and glassy state, according to the lower *T*_P3_ and higher *T*_P1_, *T*_P2_ than those of Zr0 ribbon. It is known that Fe-B can form the network in the Fe-based glasses [48]. Thus, it is expected that the corrosion resistance deduced from potential dynamic scans of the as-spun Zr0.5 and Zr1 ribbons is higher than that of the as-spun Zr0 ribbon (Figure 5). It is understood that the free volume in as-spun Zr0.5 and Zr1 ribbons is higher than the as-spun Zr0 ribbon.

With increasing *c*_Zr_, the as-spun Zr0, Zr0.5 and Zr1 amorphous ribbons have gradually increased degradation rate of MB solution (Figure 3d), indicating an increased exciting ability in Fenton-like reaction. The as-spun Zr0, Zr0.5 and Zr1 amorphous ribbons formed the cotton floc structure, coarse porous structure and 3D nano-porous structure on the ribbon surfaces during Fenton-like reaction with MB solution (Figure 4d−f), respectively. Obviously, the 3D nano-porous structure formed on the surface of Zr1 ribbon has a larger specific surface area, which can provide more reactive sites, thus improving the degradation performance of Zr1 ribbon to MB solution. According to DSC curves and electrochemical tests, the higher resistance and *T*_P1_, *T*_P2_ of Zr0.5 and Zr1 ribbon can keep the network, i.e., ligaments of the ribbons, being more stable than Zr0 ribbon. On the other hand, the $\delta_{c}$ value of reacted Zr0, Zr0.5 and Zr1 ribbons is 2.031, 2.035 and 2.039 Å, respectively, and the $\delta_{c}$ decrement values (Δ$\delta_{c}$) of Zr0, Zr0.5 and Zr1 ribbons after reaction are 0.005, 0.007 and 0.011 Å, respectively. These data indicate that the free volumes have segregated and fromed the pores on the surface of the ribbons. Hence, it is understood that the pore size increases simultaneously.

Under different *C*_H2O2_, the Zr0, Zr0.5 and Zr1 ribbons have different *η* values in MB solution and have a maximal *η* as *C*_H2O2_ = 10 mM (Figure 7d). Meanwhile, in the range of *C*_H2O2_ = 0.5–30 mM, the *η* of Zr0.5 and Zr1 ribbons is higher than Zr0 ribbon, indicating that the existence of Zr element would slow down the change of *η* in wide *C*_H2O2_ range. As *C*_H2O2_ is 1 mM, the amount and size of acicular Fe-based oxides on Zr1 ribbon surface are higher than on the Zr0 surface. Hence, the interface of Fe_x_O_y_/matrix on Zr0 ribbon surface is more compatible than on Zr1 ribbon surface, indicating that the Fe atom below the Zr1 ribbon surface can move to the outer layer more easily than on the Zr0 ribbon surface. In addition, under the help of the porous structure, the specific surface area of the Zr1 ribbon is higher and the *η* is higher than Zr0 ribbon. As *C*_H2O2_ reaches 30 mM, the amount and size of Fe_x_O_y_ are very high, and cover the pore mouth and inhibit the inner Fe atom transport to the surface to join Fenton-like reaction. This explains why the *η* of Fe-based ribbons decreases as *C*_H2O2_ increases from 30 mM.

Based on the analysis of the elemental information, surface micro-morphology and micro-structure of Zr0 and Zr1 amorphous ribbons during MB solution (*T* = 298 K, pH = 3, *C*_H2O2_ = 1 mM and *C*_MB_ = 100 mg L^−1^) degradation, the pathway of this Fenton-like reaction can be drawn, as shown schematically in Figure 9. According to $\delta_{c}$ of as-spun Zr0 and Zr1 ribbons, the free volume in Zr1 ribbon is higher Zr0 ribbon. In the reaction process, the Fe atoms can diffuse under the help of free volume and more to the surface to join the Fenton-like reaction to degrade the MB solution. At the same time, the free volume can segregate to form the pore in the reaction. Meanwhile, *T*_P1_ and *T*_P2_ of the Zr1 ribbon is higher than that of the Zr0 ribbon, which can make the network in the glassy matrix more stable. Hence, the ligament is more easily formed in the reacted Zr1 ribbon. Moreover, Fe_x_O_y_ is not compatible with ligaments and leave the channel active, which helps iron atoms to transport toward the surface from the inner part of the ribbon matrix. Hence, the Zr1 ribbon has a higher *η* and *k* than Zr0 ribbon under the same condition. These results not only prove that Zr1 amorphous ribbon has high degradability to organic dyes, but also reveal that Zr element has the role of tunnel construction in Fe-based alloys.

## 5. Conclusions

In this work, we have prepared Zr0, Zr0.5 and Zr1 amorphous ribbons with melt-spun method and we studied the microstructure, MB solution degradation behavior with several technologies. With increasing *c*_Zr_, the as-spun Zr0, Zr0.5 and Zr1 amorphous ribbons have gradually increased degradation rate of MB solution. According to $\delta_{c}$ of as-spun Zr0, Zr0.5 and Zr1 ribbons, the free volume in Zr1 ribbon is higher for Zr0 and Zr0.5 ribbons. In the reaction process, the 3D nano-porous structure formed on the surface of Zr1 ribbon has a higher specific surface area than the cotton floc structure formed by Zr0 ribbon and coarse porous structure formed by Zr0.5 ribbon. We prove that the high free volume makes it easy to form a pore structure in the reaction process, and the construction of these tunnels is beneficial to the transport of iron atoms inner the ribbon to the surface. Thus, the Zr1 ribbon has high *η* and *k* than Zr0 and Zr0.5 ribbons under the same condition. The nonmonotonic influence of pH value and H_2_O_2_ concentration on the degradation *η* and *k* of Zr1 ribbon is similar to Zr0 and Zr0.5 ribbons. This work not only provides a new method for the remediation of organic dye wastewater, but also extends the application prospect of Fe-based amorphous alloys.

## Figures and Tables

**Figure 1 materials-13-03694-f001:**
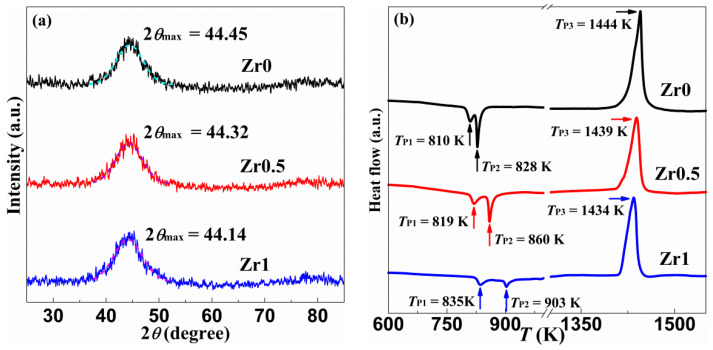
(**a**) The X-ray diffraction (XRD) curves of as-spun Zr0, Zr0.5 and Zr1 ribbons, (**b**) the differential scanning calorimetry (DSC) curves of the as-spun Zr0, Zr0.5 and Zr1 ribbons.

**Figure 2 materials-13-03694-f002:**
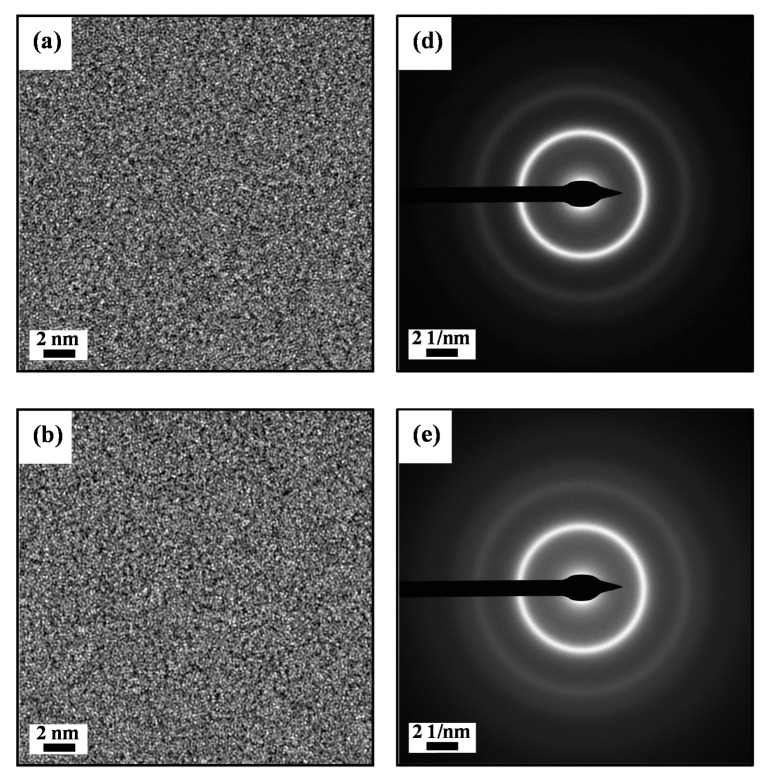
The transmission electron microscopy (TEM) images of the as-spun (**a**) Zr0, (**b**) Zr0.5 and (**c**) Zr1 ribbons, the SAED (Selected area electron diffraction)patterns of the as-spun (**d**) Zr0, (**e**) Zr0.5 and (**f**) Zr1 ribbons.

**Figure 3 materials-13-03694-f003:**
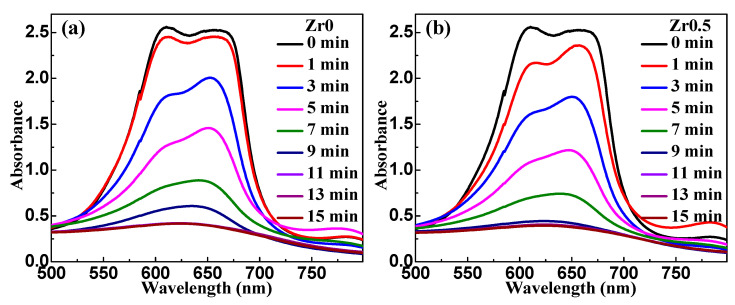
The ultraviolet–visible (UV–Vis) absorbance spectra of methylene blue (MB) solution during the Fenton-like reactions using as-spun (**a**) Zr0, (**b**) Zr0.5 and (**c**) Zr1 ribbons and (**d**) the normalized concentration change of MB solution during the degradation process. The inset in (**d**): the ln(*C*_0_/*C*_t_)-*t*_r_ curves for as-spun Zr0, Zr0.5 and Zr1 ribbons (*T* = 298 K, pH = 3, *C*_H_2_O_2__ = 1 mM, ribbon dosage = 0.5 g L^−1^ and *C*_MB_ = 100 mg L^−1^). Symbols show the experimental data while solid lines are fitting results.

**Figure 4 materials-13-03694-f004:**
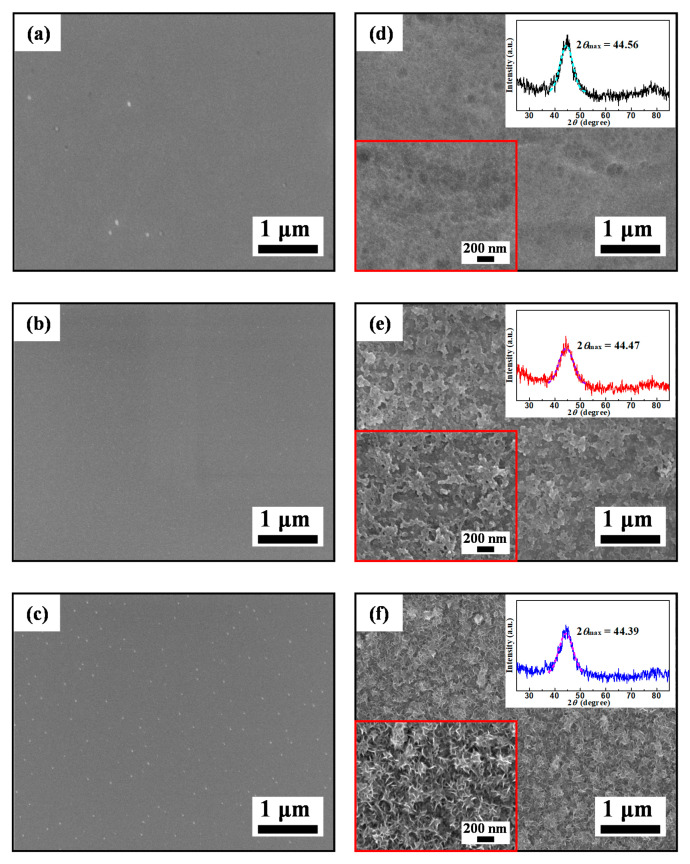
SEM micrographs of the as-spun (**a**) Zr0, (**b**) Zr0.5 and (**c**) Zr1 ribbons, and reacted (**d**) Zr0, (**e**) Zr0.5 and (**f**) Zr1 ribbons The insets in (**d**–**f**): the high-magnification images and XRD patterns of reacted Zr0, Zr0.5 and Zr1 ribbons.

**Figure 5 materials-13-03694-f005:**
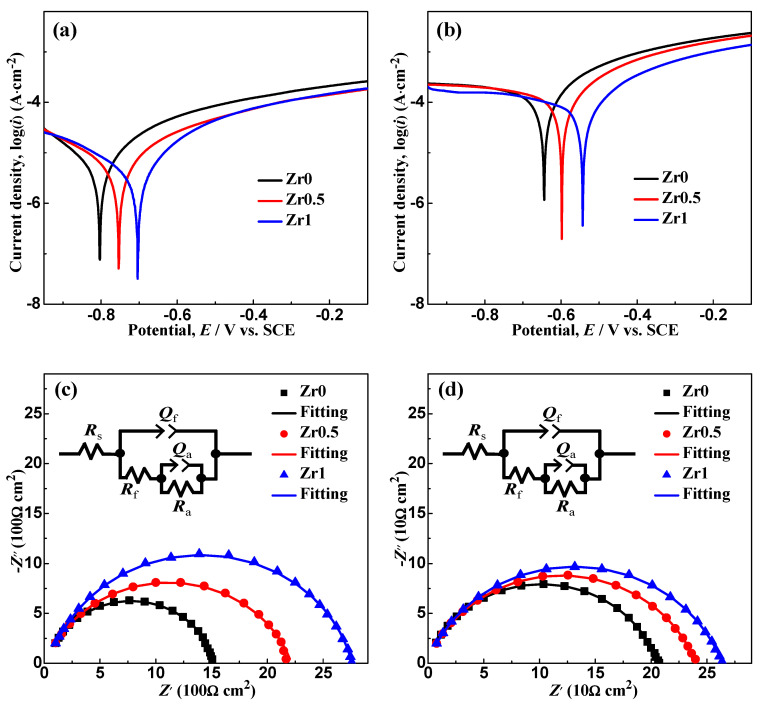
Polarization curves of the as-spun Zr0, Zr0.5 and Zr1 ribbons in (**a**) DW and (**b**) MB solution (*T* = 298 K, pH = 3, *C*_H2O2_ = 1 mM and *C*_MB_ = 100 mg L^−1^) and their Nyquist curves in (**c**) DW and (**d**) MB solution. The insets in (**c**,**d**): the general fitted circuit. Symbols show the experimental data while solid lines are fitting results.

**Figure 6 materials-13-03694-f006:**
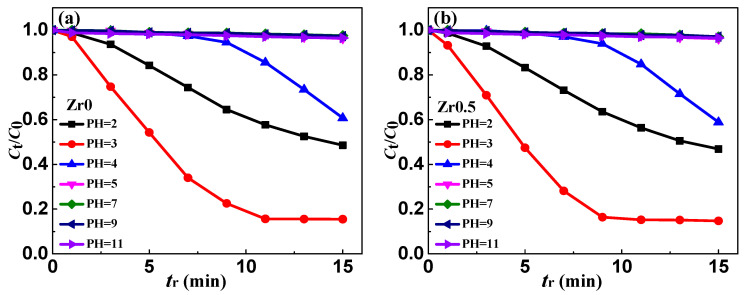
The normalized concentration *C*_t_/*C*_0_ change of MB solution during the degradation process of the as-spun (**a**) Zr0, (**b**) Zr0.5 and (**c**) Zr1 ribbons at different pH values. (**d**) The degradation efficiency (*η* = (1 − *C*_t_/*C*_0_ × 100%, *t* = 15 min) of the degradation process vs. pH for Zr0, Zr0.5 and Zr1 ribbons.

**Figure 7 materials-13-03694-f007:**
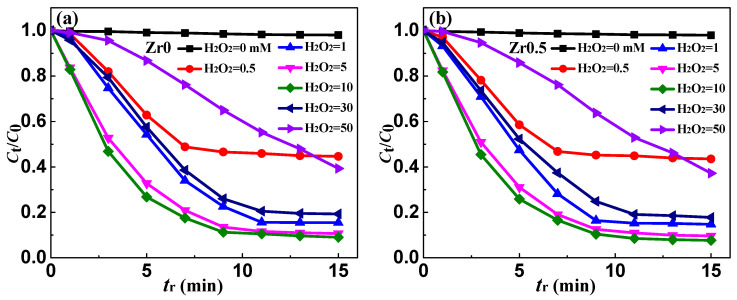
The normalized concentration *C*_t_/*C*_0_ change of MB solution during the degradation process of the as-spun (**a**) Zr0, (**b**) Zr0.5 and (**c**) Zr1 ribbons at different H_2_O_2_ concentration. (**d**) The degradation efficiency (*η* = (1 − *C*_t_/*C*_0_ × 100%, *t* = 15 min) of the degradation process vs. *C*_H_2_O_2__ for Zr0, Zr0.5 and Zr1 ribbons.

**Figure 8 materials-13-03694-f008:**
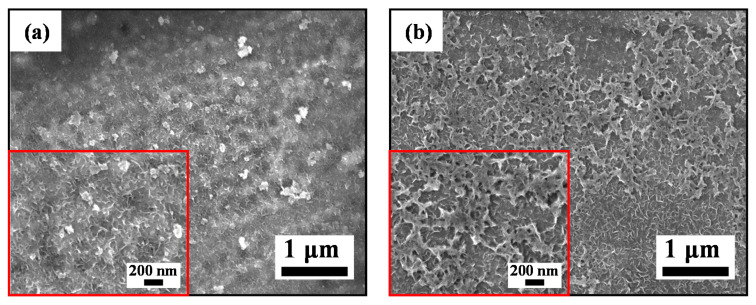
SEM micrographs of (**a**) H_2_O_2_ = 0 mM, (**b**) H_2_O_2_ = 0.5 mM, (**c**) H_2_O_2_ = 5 mM, (**d**) H_2_O_2_ = 10 mM, (**e**) H_2_O_2_ = 30 mM and (**f**) H_2_O_2_ = 50 mM for the as-spun Zr1 ribbons after reacted with MB solution. The insets in (**a**–**f**): the high-magnification images.

**Figure 9 materials-13-03694-f009:**
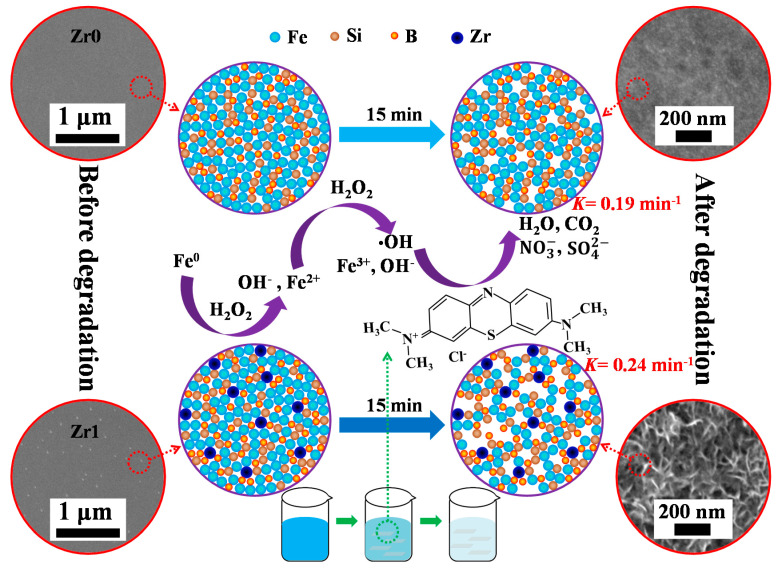
Schematic diagrams of the pathway of MB solution degradation in Fenton-like reaction using the Zr0 and Zr1 amorphous ribbons.

**Table 1 materials-13-03694-t001:** Energy dispersive X-ray spectrometer (EDS) analysis of the Zr0, Zr0.5 and Zr1 ribbons before and after reacted (at.%).

Alloy	Before Reacted					After Reacted				
	*c* _Fe_	*c* _Si_	*c* _B_	*c* _Zr_	*c* _O_	*c* _Fe_	*c* _Si_	*c* _B_	*c* _Zr_	*c* _O_
**Zr0**	77.8	8.7	10.9	-	2.6	76.3	8.6	10.5	-	4.6
**Zr0.5**	75.5	7.9	13.0	0.7	2.9	62.1	7.1	11.3	0.8	18.7
**Zr1**	76.1	9.0	10.3	1.4	3.2	53.4	8.5	9.6	1.6	26.9

**Table 2 materials-13-03694-t002:** Parameters from EIS measurements: *R*_s_, solution resistance; *Q*_f_ and *R*_f_, resistance of passivation film; *Q*_a_ and *R*_a_, resistance of electrochemical reaction; *R*_total_, total resistance.

Solution	Alloy	*R*_s_ (Ω·cm^2^)	*Q* _f_		*R*_f_ (Ω·cm^2^)	*Q* _a_		*R*_a_ (Ω·cm^2^)	*R*_total_ (Ω·cm^2^)
			*Y*_f_ (Ω^−1^·s^−n^·cm^−2^)	*N* _f_		*Y*_a_ (Ω^−1^·s^−n^·cm^−2^)	*N* _a_		
**DW**	Zr0	95.3	5.3 × 10^−9^	0.99	895.7	5.2 × 10^−9^	0.98	511.0	1502.0
	Zr0.5	125.7	3.4 × 10^−9^	0.99	1932.2	3.5 × 10^−8^	0.85	593.0	2650.9
	Zr1	147.6	7.8 × 10^−9^	0.95	2842.3	8.7 × 10^−9^	0.99	716.4	3706.3
**MO**	Zr0	18.3	4.8 × 10^−8^	0.99	211.2	3.4 × 10^−5^	0.91	28.4	257.9
	Zr0.5	21.1	3.9 × 10^−8^	0.98	250.7	3.0 × 10^−5^	0.92	33.7	305.5
	Zr1	24.5	3.8 × 10^−8^	0.99	274.7	2.5 × 10^−5^	0.93	39.3	338.5

**Table 3 materials-13-03694-t003:** EDS analysis of the Zr1 ribbon reacted with MB solution at different H_2_O_2_ concentrations (at.%).

Element	*C*_H2O2_ (mM)					
	0	0.5	5	10	30	50
***c*_Fe_**	69.7	61.1	47.2	43.9	36.4	29.5
***c*_Si_**	8.3	8.5	8.7	8.1	8.6	8.8
***c*_B_**	9.6	10.8	9.2	9.9	9.7	9.5
***c*_Zr_**	1.5	1.4	1.7	1.6	1.4	1.3
***c*_O_**	10.9	18.2	33.2	36.5	43.9	50.9

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
