# Peer review of "High MB Solution Degradation Efficiency of FeSiBZr Amorphous Ribbon with Surface Tunnels"

_materials, 2020, doi:10.3390/ma13173694_

Round 1

Reviewer 1 Report

This paper reports the degradation of methylene blue solution using (Fe78Si9B13)99.5Zr0.5 (Zr0.5) and (Fe78Si9B13)99Zr1 (Zr1) amorphous ribbons. The authors characterized the ribbons using several techniques and correlate the results. This work not only provides a new highly efficient and low-cost method for organic dyes wastewater remediation, but also extends the application fields of Fe-based alloys.. Although it contains novel and important aspects, I cannot recommend publication of this paper in the present form. However, I suggest major revision of the following points as long as the manuscript is resubmitted in the correct version:

  • General comment: The whole paper suffers from a lack of clarity and requires major editing for English in this respect. There are a number of grammatical errors and instances of badly worded/constructed sentences. The manuscript should be checked and the language refined.
  • In the abstract the authors use several abbreviations, which is not clear for the reader what means; they have to be defined before.
  • In the introduction, lines 56-58 the sentence is confused.
  • The description of the preparation of the alloy ingots it is confused. What type of pump was used? Substitute vacuumed by pumped. The authors claim that the system was pumped, and that it was filled with argon; What was the pressure inside the furnace?
  • Line 114 substitute degree by º; line 120 what the authors talk about the difference between the crystallization temperatures and or melting temperature, the used nomenclature should be specified.
  • Also in the experimental section the contact angle measurements details have to be specified.
  • All the variables in the equations (1), (2) and (3) must be defined.
  • It is not clear for me why the authors did the electrochemical analysis. Please justify?
  • When the authors present the results of pH, they write PH; that is wrong! Please correct along the manuscript
  • In the conclusions section I would like to see highlighted the effect of pH and the concentration of H2O2 on the ribbons.

Author Response

Response to Reviewer 1 Comments

Point 1: General comment: The whole paper suffers from a lack of clarity and requires major editing for English in this respect. There are a number of grammatical errors and instances of badly worded/constructed sentences. The manuscript should be checked and the language refined.

Response 1: We have carefully revised the English grammar and structure of this manuscript, which is marked with red underlines.

Point 2: In the abstract the authors use several abbreviations, which is not clear for the reader what means; they have to be defined before.

Response 2: Several abbreviations that appear in the abstract have been defined and revised in the manuscript, which is marked with red underlines.

Point 3: In the introduction, lines 56-58 the sentence is confused.

Response 3: We have revised this sentence in the manuscript, which is marked with red underlines.

Point 4: The description of the preparation of the alloy ingots it is confused. What type of pump was used? Substitute vacuumed by pumped. The authors claim that the system was pumped, and that it was filled with argon; What was the pressure inside the furnace?

Response 4: The furnace was first evacuated by the vacuum pump (ATB Serie P71 PBF 71/2B-11RQ), and then filled with argon. The atmospheric pressure inside the furnace was hold at 0.8 bar.

Point 5: Line 114 substitute degree by º; line 120 what the authors talk about the difference between the crystallization temperatures and or melting temperature, the used nomenclature should be specified.

Response 5: We have revised in the manuscript to replace degree with º of line 114. Line 120 talks about the difference between crystallization temperature and or melting temperature, and the nomenclature used is also described in the manuscript, which is marked with red underlines.

Point 6: Also in the experimental section the contact angle measurements details have to be specified. All the variables in the equations (1), (2) and (3) must be defined.

Response 6: We checked our manuscript carefully and confirmed that the “contact angle” had never been mentioned. In this article, the diffraction angle (2qmax) corresponding to the maximum diffuse scattering peak in the XRD pattern of amorphous ribbon is determined by Gaussian fitting. We have defined all the variables in equations (1), (2) and (3) in the manuscript, which is marked with red underlines.

Point 7: It is not clear for me why the authors did the electrochemical analysis. Please justify?

Response 7: The electrochemical analysis gives a quantitative description on the corrosion resistance of Zr0, Zr0.5 and Zr1 ribbons. Since the ribbon is corroded during degradation, it is important to measure the rates of mass loss of different ribbons to characterize their reusability.

Point 8: When the authors present the results of pH, they write PH; that is wrong! Please correct along the manuscript.

Response 8: We have revised them all in the manuscript.

Point 9: In the conclusions section I would like to see highlighted the effect of pH and the concentration of H2O2 on the ribbons.

Response 9: We have emphasized the effect of pH and H2O2 concentration on the degradation performance in conclusion.

Reviewer 2 Report

I was very fortunate to read and revise Your manuscript entitled “High MB solution degradation efficiency of FeSiBZr amorphous ribbon with surface tunnels” to be considered for publication in the prestigious journal, Materials. Manuscript ID: materials-893492

I hope You will find the comments thoughtful and the suggestions constructive, which can help to improve the quality of this manuscript.

The introduction of the manuscript is well written and coherent, by presenting the environmental problem of dye contaminated waters and providing a new method to help the water cleaning technology progression. It also includes relevant references; the whole manuscript contains 48 references out of which 37 (77%) were published after 2010 (22 even after 2017). The aim of the research was highlighted at the end of the introduction part moreover, as an emphasis, it was paraphrased in the conclusion section.

I suggest that the authors provide a section where they enlist the used abbreviation if it is possible.

The research design is appropriate, the authors used many, high valuable and accurate analytical methods to study the degradation efficiency of MB dye and to describe the used amorphous material, FeSiBZr.

The method section is adequately described, so the measurements and calculations can be repeatable. However, some equations used to calculate for example the kinetic process was in the results section of the manuscript. In the Methods section, authors should provide information on the used pH electrode and pH-meter and the protocol for determining the solution’s pH values.

The results were clearly presented, with the help of high-resolution graphs and valuable data. Can the authors provide information on the EDS measurements, how many parallel measurements were done?

Some spelling and editing comments:

  • frequently, the authors write PH instead of pH (for example in lines 66, 97,104,165, 202 …), this also applies for Figure 6.
  • the use of a consistent font is recommended in lines 54-55,
  • ?H2O2 should be with the same font as the rest of the text (for example in lines 104, 105, 262…), mainly in section 6 Effect of H2O2 concentration on ribbon degradation and surface morphology. This also applies for Table 3.
  • Lines 131-136 should be revised: There is mainly maze shape pattern without crystallites in the high resolution bright field images of the as-spun Zr0, Zr0.5 and Zr1 ribbons (Fig. 2(a), (b) and (c)), and the corresponding SAED patterns have two typical diffraction halos (Fig. 2(d), (e) and (f)), confirming that the as-spun Zr0, Zr0.5 and Zr1 ribbons own a fully amorphous Thus, the results of TEM are agreeing with the XRD patterns and DSC curves (Fig. 1(a) and (b)).

The conclusions were supported by the results and discussion sections.

I am confident that the  revised, newer version of the manuscript  will  be  greatly improved, therefore I suggest the publication of the manuscript “High MB solution degradation efficiency of FeSiBZr amorphous ribbon with surface tunnels” to be considered for publication in the prestigious journal, Materials after minor revision (corrections to minor methodological errors and text editing).

Author Response

Response to Reviewer 2 Comments

Point 1: The introduction of the manuscript is well written and coherent, by presenting the environmental problem of dye contaminated waters and providing a new method to help the water cleaning technology progression. It also includes relevant references; the whole manuscript contains 48 references out of which 37 (77%) were published after 2010 (22 even after 2017). The aim of the research was highlighted at the end of the introduction part moreover, as an emphasis, it was paraphrased in the conclusion section.

Response 1: Thank you.

Point 2: I suggest that the authors provide a section where they enlist the used abbreviation if it is possible.

Response 2: All abbreviations used in the manuscript were explained in detail when they are first mentioned, and we do not think it is necessary to provide an extra section to list them all.

Point 3: The research design is appropriate, the authors used many, high valuable and accurate analytical methods to study the degradation efficiency of MB dye and to describe the used amorphous material, FeSiBZr.

Response 3: Thank you.

Point 4: The method section is adequately described, so the measurements and calculations can be repeatable. However, some equations used to calculate for example the kinetic process was in the results section of the manuscript. In the Methods section, authors should provide information on the used pH electrode and pH-meter and the protocol for determining the solution’s pH values.

Response 4: The coefficients calculated by the equation are essential to the description on the following results. Regarding the determination of pH value in solution, we configured it directly through the combination of high-precision pipette and ten-thousandth bit electronic scale, without using pH electrode and pH meter.

Point 5: The results were clearly presented, with the help of high-resolution graphs and valuable data. Can the authors provide information on the EDS measurements, how many parallel measurements were done?

Response 5: The EDS results have been listed in Table 1 and Table 3 in the manuscript. We have done two parallel measurements and the parallel EDS results of as-spun Zr0, Zr0.5 and Zr1 ribbons are given in the following figure.

First measurement                         Second measurement

As-spun Zr0 ribbon

As-spun Zr0.5 ribbon

As-spun Zr1ribbon

Point 6: Some spelling and editing comments:

  • frequently, the authors write PH instead of pH (for example in lines 66, 97,104,165, 202 …), this also applies for Figure 6.
  • the use of a consistent font is recommended in lines 54-55,
  • ?H2O2 should be with the same font as the rest of the text (for example in lines 104, 105, 262…), mainly in section 3.6 Effect of H2O2 concentration on ribbon degradation and surface morphology. This also applies for Table 3.
  • Lines 131-136 should be revised: There is mainly maze shape pattern without crystallites in the high resolution bright field images of the as-spun Zr0, Zr0.5 and Zr1 ribbons (Fig. 2(a), (b) and (c)), and the corresponding SAED patterns have two typical diffraction halos (Fig. 2(d), (e) and (f)), confirming that the as-spun Zr0, Zr0.5 and Zr1 ribbons own a fully amorphous Thus, the results of TEM are agreeing with the XRD patterns and DSC curves (Fig. 1(a) and (b)).

Response 6: The above spelling and editing comments have been revised in the manuscript, which is marked with red underlines.

Point 7: The conclusions were supported by the results and discussion sections.

Response 7: Thank you.

Round 2

Reviewer 1 Report

The authors improved the manuscript in the revised version and considered all the suggestions. I suggest to accept after some minor revisions:

Line 369: where it is "with varying technology" I suggest "with several technology"

Line 372: where it is: free volume in Zr1 ribbon is higher Zr0 and Zr0.5 ribbons. In reaction process" I suggest "free volume in Zr1 ribbon is higher  for Zr0 and Zr0.5 ribbons. In the reaction process"

Author Response

Response to Reviewer 1 Comments

Point 1: Line 369: where it is "with varying technology" I suggest "with several technology"

Response 1: We have revised this sentence in the manuscript, which is marked with red underlines.

Point 2: Line 372: where it is: free volume in Zr1 ribbon is higher Zr0 and Zr0.5 ribbons. In reaction process" I suggest "free volume in Zr1 ribbon is higher for Zr0 and Zr0.5 ribbons. In the reaction process"

Response 2: We have revised this sentence in the manuscript, which is marked with red underlines.